# Nodal Spread Prediction in Human Oral Tongue Squamous Cell Carcinoma Using a Cancer-Testis Antigen Genes Signature

**DOI:** 10.3390/ijms26189258

**Published:** 2025-09-22

**Authors:** Yoav Smith, Amit Cohen, Tzahi Neuman, Yoram Fleissig, Nir Hirshoren

**Affiliations:** 1Genomic Data Analysis Unit, Faculty of Medicine, The Hebrew University of Jerusalem, Jerusalem 9112102, Israel; 2Department of Pathology, Hadassah Medical Center, Jerusalem 91120, Israel; 3Faculty of Medicine, The Hebrew University of Jerusalem, Jerusalem 91120, Israel; 4Department of Oral and Maxillofacial Surgery, Hadassah Medical Center, Faculty of Dental Medicine, Hebrew University of Jerusalem, Jerusalem 91120, Israel; 5Department of Otolaryngology/Head & Neck Surgery, Hadassah Medical Center, Jerusalem 91120, Israel

**Keywords:** oral tongue squamous cell carcinoma, cancer-testis antigens (CTA), neck lymph node metastasis, NanoString nCounter, machine learning, biomarker

## Abstract

Cervical lymph node metastasis is the strongest prognostic factor in oral tongue carcinoma, yet current clinical guidelines rely primarily on depth of invasion to guide elective neck dissection. This approach results in unnecessary surgery in up to 70% of patients. Cancer-testis antigens (CTAs) are a family of genes associated with tumor aggressiveness and may serve as predictive biomarkers for nodal spread. A multi-step analysis integrating large-scale public datasets, including microarray (GSE78060), bulk RNA-seq emerging from the cancer genome atlas (TCGA), and single-cell RNA-seq (GSE103322), was employed to identify CTA genes active in oral tongue cancer. Selected genes were validated using NanoString nCounter RNA profiling of 16 patients undergoing curative glossectomy with elective neck dissection. Machine learning algorithms, including decision trees, t-distributed stochastic neighbor embedding (t-SNE), and convolutional neural networks (CNN), were applied to assess predictive power for nodal metastasis. Computational analysis initially identified 40 cancer-active CTA genes, of which four genes (LY6K, MAGEA3, CEP55, and ATAD2) were most indicative of nodal spread. In our patient cohort, NanoString nCounter profiling combined with machine learning confirmed these four genes as highly predictive. We present a proof-of-concept CTA-based genetic diagnostic tool capable of discriminating nodal involvement in oral tongue cancer. This approach may reduce unnecessary neck dissections, minimizing surgical morbidity.

## 1. Introduction

Head and neck cancer (HNC) represents a significant health concern globally. It is the seventh most common cancer, with approximately 890,000 new cases diagnosed each year, reflecting about 3% of all cancers in the United States. The incidence rates of HNC have been steadily increasing in the U.S., highlighting a growing public health issue [1,2].

Oral cancer is particularly notable, as it ranks as the eighth most commonly diagnosed cancer among males worldwide. Within this category, the tongue is identified as the most frequently affected site for oral cavity carcinoma, and the incidence of this has also been on the rise [3,4].

The predominant histological type of oral cancer is squamous cell carcinoma (SCC), which accounts for over 90% of oral cancer cases [5]. Despite advancements in treatment, the overall 5-year survival rate for patients diagnosed with oral cancer remains around 50%, underscoring the need for increased awareness, early detection, and improved therapeutic strategies [6].

The management of early-stage (T1, T2) tongue squamous cell carcinoma (SCC) without clinical evidence of nodal involvement (N0) primarily involves the surgical excision of the primary tumor with sufficient oncological margins, which may involve various types of glossectomy. Additionally, elective neck dissection (END) is often performed to identify and remove any microscopic nodal metastases. The current clinical standard suggests that a depth of invasion (DOI) of 3 mm or more is a key indicator for recommending elective neck dissection, reflecting the concern for potential nodal involvement [7]. However, it is reported that approximately 70% of patients who undergo elective neck dissection with clinically negative nodes remain pathologically negative for nodal disease, indicating that these patients may be overtreated. In contrast, the remaining 30% who do have nodal metastases could benefit significantly from the procedure, as the removal of microscopic nodal disease is associated with improved survival outcomes and may alter the course of adjuvant treatments available [6,8,9].

Despite the survival advantages for those with nodal involvement, the practice of elective neck dissection raises concerns related to patient quality of life, potential complications arising from the surgery [10], and financial and time burdens associated with unnecessary procedures [11]. These factors highlight the need for more precise methods to identify patients who would genuinely benefit from neck dissection, thus reducing overtreatment and its associated risks while maintaining beneficial outcomes for those with nodal metastases.

Cancer-testis antigens (CTAs) are a unique class of proteins predominantly expressed in the testes and placental tissue, but aberrantly expressed in a variety of malignancies [12,13]. Their association with the diagnosis, development, and treatment of solid tumors [14,15] prompted us to investigate the expression patterns of the CTA gene family in oral tongue carcinoma (see Patent 12,018,331 by Smith Y [16]). By examining the correlation between CTA expression and pathological outcomes—specifically whether cervical lymph nodes remain free of malignancy—we aim to identify potential biomarkers that could help predict the risk of occult metastases in affected patients. 

The CTA dataset comprises more than 200 genes (GeoMx^®^ Cancer Transcriptome Atlas), some located on the X chromosome—such as MAGE, GAGE, XAGE, SPANX, and CT45—and others located on various autosomes, including CEP55, TTK, PBK, ATAD2, CTNNA2, and LY6K. CTA genes are categorized into three biologically meaningful groups: 1. Non-X-linked CTAs. 2. X-linked CTAs. 3. Non-X-linked CTAs involved in cell proliferation or division.

In our study, we employed various computational approaches to investigate CTA genes that are active in cancers, including head and neck cancers, using a large, previously published dataset comprising CTA gene expression profiles from both cancerous and healthy tissues. Genes exhibiting minimal dynamic changes were excluded, allowing us to focus on those with significant activity. We then analyzed these active cancer-related CTA genes using RNA extracted from primary tongue carcinoma samples collected by our research group. This analysis identified a smaller subset of highly active, selectively expressed genes that appear to be particularly relevant in tongue cancer and may serve as predictors of cervical lymph node metastasis.

To validate our selected CTA genes and assess their predictive value for nodal spread, we applied machine learning algorithms to biopsy data from patients with oral tongue carcinoma, encompassing a range of nodal statuses (N0, N1, N2). These tools enabled us to evaluate the accuracy of our gene-based predictions and their potential clinical utility. Genomic analysis of tongue biopsies performed prior to treatment decisions may address a critical unmet need in clinical practice. Our approach aims to reduce unnecessary diagnostic surgeries by identifying patients who are unlikely to benefit from neck surgery, while accurately selecting those who are most likely to benefit from surgical intervention.

## 2. Results

### 2.1. Patient Cohort and Clinical Characteristics (Table 1)

Of the 20 patients initially enrolled, 16 (80%) were included in the final analysis due to RNA extraction failure in 4 cases. The study cohort comprised 62.5% females, with a mean age of 60.6 years (median: 64.5; range: 24–77 years). The mean follow-up duration was 36.6 months.

All patients underwent curative-intent glossectomy combined with elective neck dissection. Pathological evaluation revealed positive lymph node involvement in nine patients (56.3%), of whom five (55.5%) exhibited extra-nodal extension (ENE). Advanced disease (stage III or IV) was present in 10 patients (62.5%).

Regarding adjuvant therapy, 12 patients (63.1%) received postoperative radiotherapy, and 7 of these (43.8%) also received concomitant chemotherapy. During the follow-up period, 4 patients (25%) died of disease, with an average time to death of 12 months.

### 2.2. Genomic Computational Analysis

We compared the expression profiles of 207 cancer-testis antigen (CTA) genes across 166 different cancer types (2155 subjects; GEO: GSE2109, Expression Project for Oncology [EXPO], Public on 15 January 2005) to those from non-cancerous tissues (449 subjects; GEO: GSE7307, Public on 9 April 2007, Human Body Index-excluding testis and trophoblast tissues). Both datasets were generated using the same Affymetrix Human Genome U133 Plus 2.0 Array platform. Prior to analysis, all expression values were normalized by scaling each column to a 0–1 range.

Analysis revealed heterogeneous CTA expression patterns across different cancer types. We identified 40 CTA genes as cancer-associated, demonstrating expression levels at least twofold higher than the normal tissue average. Volcano plot analysis of these 40 genes enabled their classification into three distinct subgroups based on chromosomal location, expression consistency, and functional correlation (Figure 1).

Subgroup 1: X-chromosome CTA genes that were randomly distributed in terms of expression level, either low or highly expressed, and not strongly correlated with other genes. These appeared on the far right of the X-axis in the volcano plot (indicating substantial upregulation relative to normal) but low on the Y-axis, reflecting their stochastic expression pattern (depicted in black).

Subgroup 2: Non-X-chromosome CTA genes associated with the mitotic cell cycle. These genes consistently appeared across multiple cancer types and exhibited strong inter-gene correlations. Their consistent presence and correlation resulted in high Y-axis positions, while their X-axis position corresponded to their degree of overexpression (depicted in red).

Subgroup 3: Non-X-chromosome CTA genes unrelated to the mitotic cell cycle, representing a distinct expression pattern (depicted in green).

### 2.3. Focused Computational Analysis on Tongue Cancer Datasets

Subsequent computational analysis was restricted to datasets comprising exclusively tongue cancer cases. This refinement identified four highly selected CTA genes, all of which were later confirmed by our wet-lab NanoString nCounter analysis. These genes were further categorized into three biologically relevant groups based on gene correlation analysis: non-X-linked: LY6K; X-linked: MAGEA3; proliferation/cell division-related: CEP55 and ATAD2.

The expression profiles and correlation patterns of these genes were consistently observed across multiple datasets. Specifically, the analysis of GSE78060 (microarray data) is presented in Figure 2, bulk RNA sequencing data from TCGA in Figure 3, and single-cell RNA sequencing data from GSE103322 in Figure 4.

### 2.4. NanoString nCounter Validation and CTA Gene Expression in Oral Tongue Cancer

Subsequent analyses focused on the NanoString nCounter data generated from RNA extracted from 16 oral tongue cancer patients in our cohort. Of the 40 CTA genes previously identified via computational analysis, the 36 most cancer-active genes were selected for expression comparison against neck nodal status (N0 versus N1&2). We have excluded the least cancer-associated CTA genes: Magea12, Magea9b, Oip5 and Casc5.

Normalization was performed using the NanoString nCounter internal quality control probes for each patient. Gene expression counts were compared between tumor tissue and spatially separated adjacent normal tissue to account for baseline expression differences. The resulting differential expression patterns and the ROC curves for the four selected genes are illustrated in Figure 5.

### 2.5. Machine Learning-Based Prediction of Neck Nodal Spread

To validate our selection of cancer-active CTA genes and assess their predictive power for neck nodal metastasis, we applied machine learning approaches to the NanoString nCounter RNA expression data from the 16 oral tongue cancer patients. Each of the 36 selected CTA genes was evaluated for its contribution to nodal status prediction. Gene weights were first analyzed using the Kruskal–Wallis (H) test, a non-parametric method, to identify genes significantly associated with lymph node involvement.

Further visualization (Figure 6A) and prediction (Figure 6B) models were performed using MATLAB 2024B’s applications. Across multiple analyses, including decision tree models, t-distributed stochastic neighbor embedding (t-SNE) for nonlinear dimensionality reduction and visualization (Figure 6A), and convolutional neural networks (CNN) (Figure 6B), the same four CTA genes—LY6K, ATAD2, CEP55, and MAGEA3—emerged as the most informative for nodal spread.

Using a training set of 12 (70%) patients (7 patients with nodal involvement and 5 patients with nodal negative disease) and a testing set of 4 (30%) patients (2 patients with and 2 patients without nodal spread), the CNN model achieved 100% predictive accuracy. The accuracy dropped to 81.25% (13 patients out of 16 patients) when applying the Leave-One-Out Cross-Validation (LOOCV) method, which presented a probable overfitting feature.

The combined decision tree approaches reached 93.8% accuracy. These results confirm the high predictive potential of these four CTA genes for lymphatic metastasis in oral tongue cancer.

## 3. Discussion

Cervical lymph node metastasis remains the most important prognostic factor in oral cancer, as survival is reduced by approximately 50% when nodal spread occurs, often necessitating adjuvant treatment and more intensive follow-up [17,18]. Current clinical guidelines primarily rely on a single histopathological parameter—depth of invasion—to guide the decision for elective neck dissection, despite the fact that nearly 70% of such procedures reveal no nodal involvement [9]. Neck dissections carry inherent operative risks [19] that can significantly impact patients’ quality of life, in addition to imposing substantial financial and logistical burdens on healthcare systems.

Integrating wet-lab genomic analyses with advanced in silico computational approaches offers a robust strategy for identifying novel cancer biomarkers [20]. The use of public genomic databases in conjunction with data analysis tools enables efficient screening and comparison of gene expression profiles across patient cohorts, thereby enhancing the accuracy of disease-associated gene identification.

Cancer-testis antigens (CTAs) represent a genomic family with complex, multifaceted associations with tumor aggressiveness [6] and may serve as potential biomarkers for predicting tumor spread [7]. In this study, we developed a novel, multi-step approach to establish a reliable tool for discriminating between pathologically positive and negative neck lymph nodes in oral cavity carcinoma, based on the expression of four CTA genes.

Initially, we performed large-scale computational analyses using diverse datasets, identifying 40 of the most cancer-active CTA genes. Among these, four genes (LY6K, ATAD2, CEP55, and MAGEA3) emerged as the most predictive of nodal spread specifically in tongue cancer. These findings were consistently observed across multiple platforms, including microarray, bulk RNA sequencing, and single-cell analyses. Validation in our cohort of 16 patients using NanoString nCounter assays and machine learning techniques demonstrated high predictive accuracy for these four genes.

Our findings suggest that this CTA-based genetic diagnostic tool could potentially reduce unnecessary diagnostic neck dissections, sparing patients from surgical morbidity and healthcare systems from avoidable costs.

Co-expression of CTAs LY6K, ATAD2, CEP55, and MAGEA3 further enhances oncogenic signaling, cell cycle progression, and immune escape. ATAD2 [21,22,23] and CEP55 [24,25,26] are key regulators of chromatin remodeling and cytokinesis, respectively, while MAGEA3 [27,28,29] modulates apoptosis and therapy resistance. LY6K [30,31,32] is aberrantly expressed in a wide range of solid tumors and is associated with poor prognosis. Mechanistically, LY6K activates the ERK-AKT and TGF-β pathways, which are critical for tumor cell proliferation, migration, invasion, and metastatic spread. The combined activity of these CTAs contributes to aggressive tumor behavior and resistance to conventional therapies. Thus, the impact we have witnessed and presented here in the case of their role in nodal spread prediction in human oral tongue squamous cell carcinoma might be expanded to other cancers and suggests that this potential should be studied further. This study represents a proof-of-concept investigation. Future research should include larger-scale, multicenter cohorts with blinded validation (in contrast to our investigation) and diverse patient populations to confirm the predictive utility of these CTA genes in clinical practice.

## 4. Materials and Methods

This study is a proof of concept investigation of oral tongue HNC patients treated in a single tertiary referral medical center. The study was approved by the local Internal Review Board (0198-14-HMO). All patients signed an informed consent form following a detailed explanation. Data anonymization was rigorously maintained.

### 4.1. Primary Endpoints

Analysis of the prediction power of CTA gene signatures for neck nodal spread among clinical negative oral cavity carcinoma undergoing lymph node removal (neck dissection) as indicated by current treatment guidelines.

### 4.2. Patients

Patients with clinically negative neck nodal spread of primary oral tongue SCC who were operated on between 2019 and 2022 (minimal follow-up of 36 months).

### 4.3. Main Inclusion Criteria

Patients with tongue squamous cell carcinoma (SCC) presenting clinically negative nodal involvement (based on imaging or physical examination), undergoing curative primary glossectomy with elective neck dissection, depth of invasion (DOI) ≥ 3 mm. Minimum follow-up period of 36 months after surgery to exclude cases of late nodal metastasis.

### 4.4. Exclusion Criteria

Lack of signed informed consent, immunocompromised status, insufficient surgical margins (≤5 mm following glossectomy). Patients with neoadjuvant therapy prior to surgery.

### 4.5. RNA Extraction Quality Control

RNA integrity was assessed by nucleic acid fragmentation length. Acceptance threshold: at least 50% of fragments with a length ≥ 300 nucleotides.

### 4.6. Demographic and Clinical Data

Demographic and clinical data collected included patients’ age, sex, and immunodeficiency status, whether drug-induced or disease-induced. Tumor staging was recorded both clinically and pathologically for the T and N classifications, along with the overall TNM stage according to the American Joint Committee on Cancer (AJCC) 8th edition [33]. For analysis, TNM stages were categorized as early (stages I and II) or advanced (stages III and IV). Pathological details documented included the status of surgical margins, the number and size of involved lymph nodes, and the presence of extra-nodal extension. Data regarding adjuvant treatment following surgery were also collected, specifically on the administration of postoperative radiotherapy (PORT) or chemoradiotherapy (CRT), noting that all treatment decisions were determined in multidisciplinary tumor board meetings. Survival-related information included the last follow-up date, disease status at that time (with or without evidence of disease), vital status (alive or deceased), and, for deceased patients, whether death was disease-related or due to other causes.

### 4.7. Genomic Computational Analysis

Genomic computational analysis involved the investigation of open-source datasets to examine the expression profiles of known cancer-testis antigen (CTA) genes in both healthy and cancerous tissues. This analysis utilized multiple platforms, including microarray data (GSE78060, Public on 31 July 2017), bulk RNA sequencing data from The Cancer Genome Atlas (TCGA) [34], and single-cell RNA sequencing data (GSE103322, Public on 30 November 2017) [35]. Based on these datasets, we identified a subset of actively expressed CTA genes in cancerous tissue, which served as the basis for the design and implementation of the NanoString nCounter panel used in the subsequent analysis.

### 4.8. Tissue Analysis

Oral tongue cancerous paraffin-embedded biopsies were analyzed by a pathologist expert (T.N) who marked areas of discrete interest-healthy and cancerous regions of the same patient, followed by a de-paraffinization process. This meticulous step allowed a separate (cancer) and complementary (cancer vs. healthy) genomic analysis of the primary tongue tumor for each case for best nodal spread stratification.

### 4.9. RNA Extraction

Total RNA was extracted from tongue Formalin-Fixed Paraffin-Embedded (FFPE) samples using an FFPET RNA Isolation Kit (Roche, Diagnostics GmbH, Mannheim, Germany) according to the manufacturer’s instructions. Total RNA was diluted in RNase free water (Sigma-Aldrich, Rehovot, Israel). The purity and quality of extracted RNA were evaluated using an Agilent TapStation 4200 system (Agilent Technologies, Santa Clara, CA, USA).

### 4.10. NanoString nCounter Analysis

NanoString nCounter analysis was performed using 200 ng of total extracted RNA per sample, processed on the NanoString nCounter Analysis System (NanoString Technologies, Seattle, WA, USA). Hybridization reactions were carried out according to the manufacturer’s protocol. An nCounter CodeSet (NanoString Technologies), consisting of biotinylated capture probes for target genes and reporter probes linked to color-barcode tags, was hybridized with 200 ng of total RNA for 18 h at 65 °C. Following hybridization, samples were processed using an automated nCounter Prep Station, where they were purified and immobilized in a sample cartridge for data acquisition. Quantification of target mRNA molecules was then performed using the nCounter Digital Analyzer (proprietary kit).

The resulting expression counts were analyzed using the nSolver 4.0 analysis software (NanoString Technologies), with normalization based on the geometric mean of the five standard positive controls provided by the kit. This analysis generated raw expression data for oral tongue cancer samples, focusing on the subset of cancer-testis antigen (CTA) genes previously identified as relevant through computational big-data analysis.

### 4.11. Oral Tongue Cancer CTA Gene Analysis, Expression and Accuracy

The raw expression data obtained from the NanoString nCounter RNA analysis were processed using multiple analytical approaches, including volcano plot analysis, multi-layer decision tree modeling, and a convolutional neural network (CNN) framework. The CNN architecture applied weighted inputs to generate predictive outputs, specifically classifying nodal status as either positive (with lymphatic spread) or negative (without lymphatic spread) in oral cavity cancer cases. This neural network underwent a deep learning optimization of weights to achieve the best predictive performance.


## Figures and Tables

**Figure 1 ijms-26-09258-f001:**
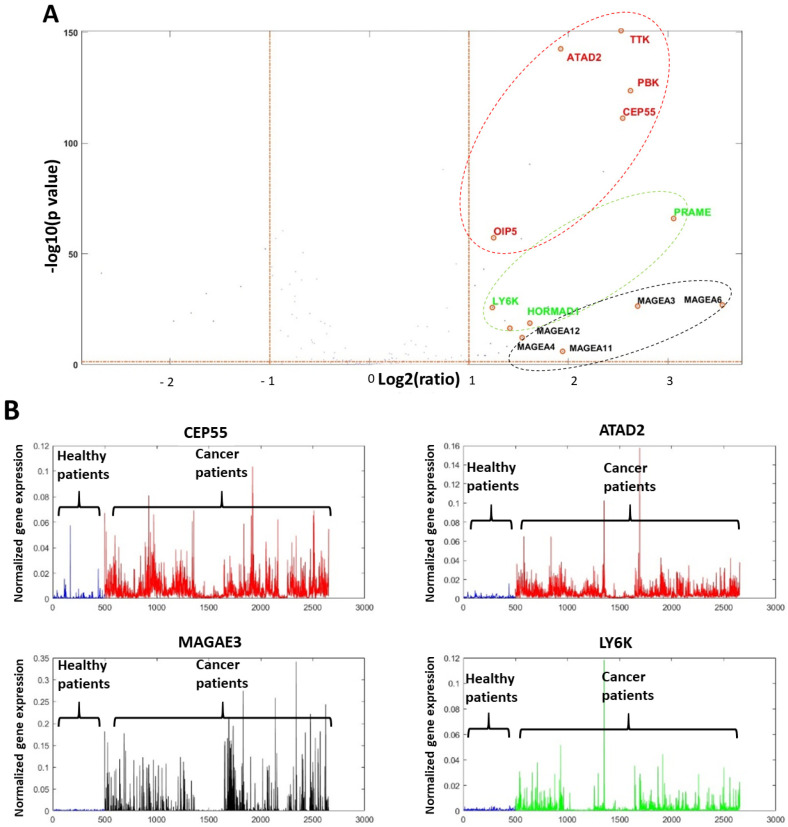
CTA gene volcano analysis presentation demonstrating the three CTA gene subgroups in 2155 cancers vs. 449 non-cancer tissues. (**A**): Volcano analysis plot. (**B**): Four main CTA most active cancer genes. Black color—X chromosome genes. Red color—mitotic and cell cycle CTA genes. Green color—other CTA genes.

**Figure 2 ijms-26-09258-f002:**
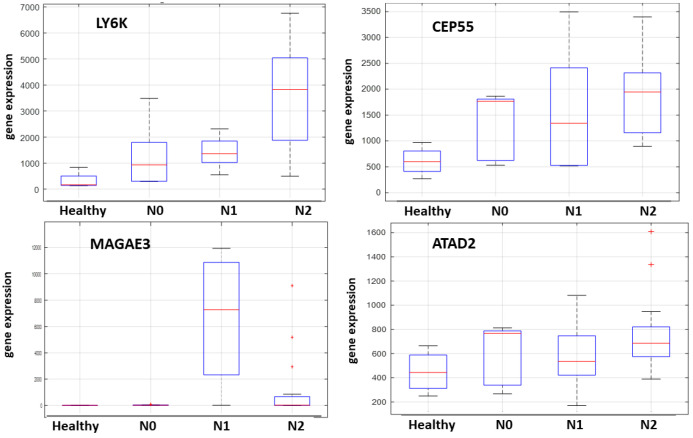
RNA expression of the four highly selected CTA genes (LY6K, ATAD2, CEP55 and MAGEA3) associated with tongue cancer nodal status emerging from the GSE78060 dataset.

**Figure 3 ijms-26-09258-f003:**
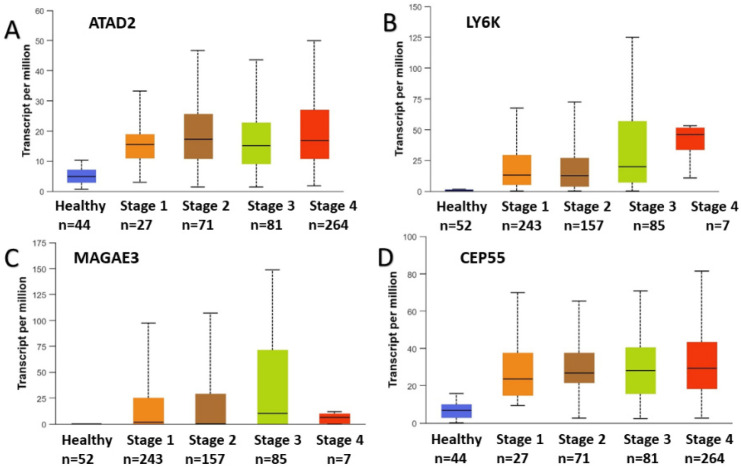
RNA expression of the selected four CTA genes: (**A**): ATAD2, (**B**): LY6K, (**C**): MAGEA3, (**D**): CEP55 which directly linked to the increase in head and neck cancer stages in the large scale DeepSeq dataset.

**Figure 4 ijms-26-09258-f004:**
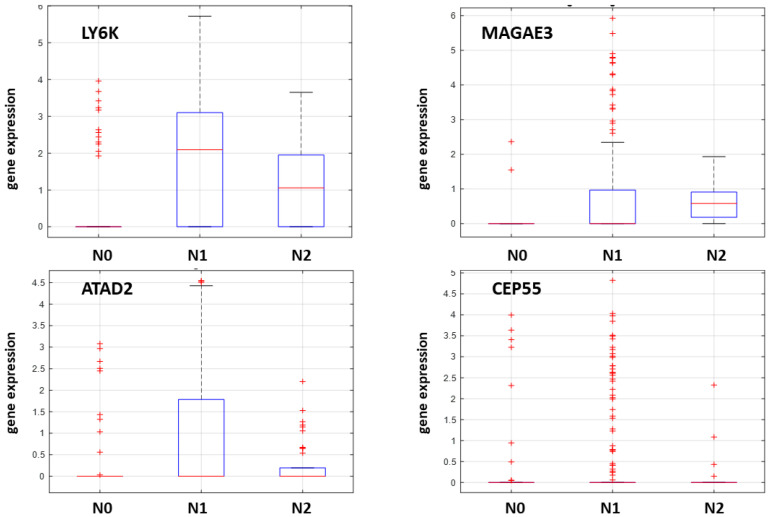
Single cell dataset (GSE103322) analysis demonstrates the RNA expression of the four highly selected CTA genes (LY6K, ATAD2, CEP55 and MAGEA3) associated with neck nodal status.

**Figure 5 ijms-26-09258-f005:**
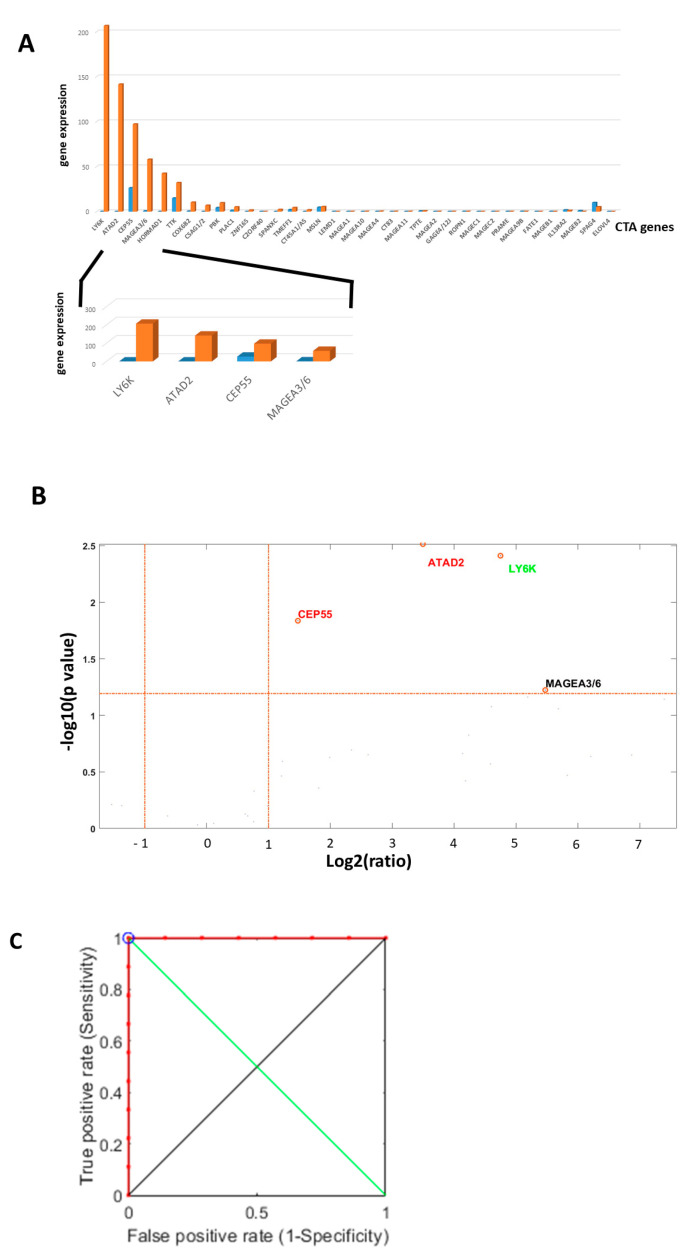
NanoString nCounter CTA gene expression in oral tongue cancer in 16 patients. (**A**): Median RNA expression of each of the 36 CTA genes measured with the NanoString nCounter kit of our 16 tongue cancer patients. We highlighted the four most profound active CTA genes. The orange-colored columns present patients with neck nodal spread, as compared to the blue-colored columns with no diseased nodal spread. (**B**): The volcano diagram, with the four genes marked directed to the right and top corner. Those four CTA genes were highly predictive for neck nodal spread. (**C**): The four selected CTA genes: ROC curve (red colored); random classifier colored black; cut-off point colored blue.

**Figure 6 ijms-26-09258-f006:**
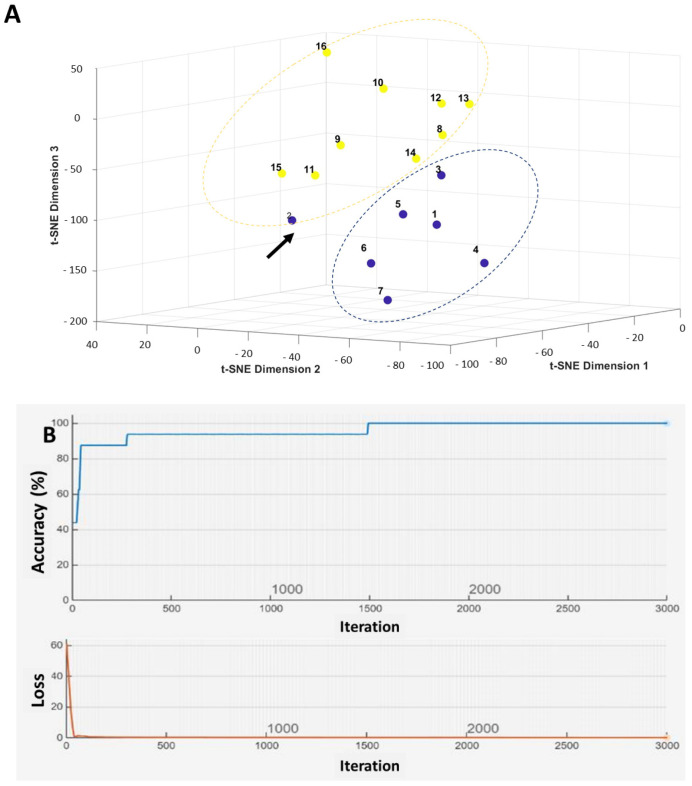
CTA gene machine learning analysis investigating 16 oral tongue carcinoma patients with the four highly selected CTA genes (LY6K, ATAD2, CEP55 and MAGEA3). (**A**): The t-distributed stochastic neighbor embedding implicating the patients with no nodal disease (blue color) and those with neck nodal spread (yellow color). Arrow marks the patients with nodal spread not detected by our CTA signature, implying 93.8% accuracy. (**B**): Convolutional neural nets (CNN) of the four highly selected CTA genes in 16 tongue cancer patients, detecting those with neck nodal spread, demonstrating 100% accuracy.

**Table 1 ijms-26-09258-t001:** Demographic and clinical data of 16 oral cavity, head and neck cancer patients.

Scheme	Female N (%)	10 (62.5%)			
	Male N (%)	6 (37.5%)			
Age, years	Mean	60.6			
	Median	64.5			
	Range	24–77			
Disease stage	Early (stage I&II) N (%)	6 (37.5%)			
	Advanced (stage III&IV) N (%)	10 (62.5%)			
Nodal disease	Positive N (%)	9 (56.25%)	Positive ENE		5 (55.5%)
			Negative ENE		4 (44.5%)
	Negative N (%)	7 (43.75%)			
Adjuvant treatment	Radiation N (%)	12 (63.15%)			
	Chemotherapy N (%)	7 (43.75%)			
Survival	Death of disease N (%)	4 (25%)	Time, months	Mean	12
				Median	12.5
				Range	8–15

ENE: Extra-nodal extension.

## Data Availability

The data are available on request from the corresponding author.

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
