# Peer review of "Nodal Spread Prediction in Human Oral Tongue Squamous Cell Carcinoma Using a Cancer-Testis Antigen Genes Signature"

_ijms, 2025, doi:10.3390/ijms26189258_

Round 1

Reviewer 1 Report

Comments and Suggestions for Authors

In this work, the authors mined public omics datasets (microarray, TCGA bulk RNA-seq, single-cell RNA-seq) to shortlist cancer-testis antigens associated with oral tongue cancer.
They then profiled 36 CTA genes in FFPE biopsies from 16 patients using NanoString and identified four genes (LY6K, MAGEA3, CEP55, ATAD2) linked to nodal metastasis.
Finally, they applied ML models to these data to predict neck nodal spread and reported high accuracy in this small proof-of-concept cohort. This work is good and reports a nice effort to collect and utilize publicly available data. The manuscript is written well. I have a few suggestions and comments.

CNN reports 100% accuracy on a 4-patient test set (L210–213); this is not convincing and suggests severe overfitting. I suggest using cross-validation/LOOCV.

Clarify whether feature selection and normalization were performed inside each CV fold; the current description implies possible data leakage from the full cohort.

t-SNE is a visualization tool, not a classifier; avoid framing Figure 6A as evidence of predictive performance. Provide ROC/PR curves, AUC, and confusion matrices.

Provide class counts (N0 vs N+) in train/test splits and use stratified CV.

The 36 NanoString genes were chosen from 40 CTAs identified on public data. I suggest that authors to specify a priori selection rule to avoid cherry-picking and list the 4 excluded genes.

Several figure legends (“.” placeholders) are incomplete; axes, units, cohort sizes, and statistics (p values, corrections) are missing in Figs 2–6.

Reviewer 2 Report

Comments and Suggestions for Authors

This manuscript presents a well-structured and carefully executed study addressing a clinically significant issue in oral tongue squamous cell carcinoma: the prediction of nodal metastasis. By integrating multiple public datasets (microarray, bulk RNA-seq, single-cell RNA-seq) and validating findings with NanoString profiling in a patient cohort, the authors convincingly identify four cancer-testis antigens (LY6K, MAGEA3, CEP55, ATAD2) as strong predictors of nodal spread. The inclusion of machine learning approaches adds further robustness to the analysis, demonstrating high predictive accuracy. The study is novel, methodologically sound, and clinically relevant, particularly as it offers a potential strategy to reduce unnecessary elective neck dissections and associated morbidity. While the sample size is relatively small, the work provides a compelling proof-of-concept with potential translational value. Overall, the manuscript is clearly written and scientifically rigorous, and I recommend acceptance after minor revisions to improve clarity and consistency.

Minor points

Introduction (page 2, line 59-62): Add a reference to support the statement “However, it is reported that approximately 70% of patients who undergo elective neck dissection with clinically negative nodes remain pathologically negative for nodal disease, indicating that these patients may be overtreated.”

Discussion (page 10, line 250): “Aggressive” may be corrected to “aggressive”.

Methods (page 12, line 339-340): It may be better to highlight the lack of blinding again in the Discussion as a study limitation.

References (page 13): Please correct duplicate numbering at Reference #1 (Siegel et al. vs Ludwig et al.).

References (page 13): The citation #6 (Levy M, Read SE. Erythema infectiosum and pregnancy-related complications) appears unrelated to this study. Please check.
